# Metabolites Potentially Determine the High Antioxidant Properties of *Limosilactobacillus fermentum* U-21

**DOI:** 10.3390/biotech12020039

**Published:** 2023-05-17

**Authors:** Yelena V. Grishina, Aleksey A. Vatlin, Dilara A. Mavletova, Maya V. Odorskaya, Alexey M. Senkovenko, Rustem A. Ilyasov, Valeriy N. Danilenko

**Affiliations:** 1Laboratory of Bacterial Genetics, Vavilov Institute of General Genetics, Russian Academy of Sciences, 119333 Moscow, Russia; grishina.e@phystech.edu (Y.V.G.); vatlin_alexey123@mail.ru (A.A.V.); lmavletova@mail.ru (D.A.M.); maya_epifanova@mail.ru (M.V.O.); valerid@vigg.ru (V.N.D.); 2Moscow Institute of Physics and Technology, National Research University, 141701 Moscow, Russia; 3Department of Bioengineering, Faculty of Biology, M.V. Lomonosov Moscow State University, Leninskie Gory 1/12, 111234 Moscow, Russia; senkoven99@gmail.com; 4Laboratory of Molecular Genetics, Bashkir State Agrarian University, 450001 Ufa, Russia

**Keywords:** *Limosilactobacillus*, antioxidant, immunomodulatory activity, gut microbiota, GC-MS, metabolomics

## Abstract

Many kinds of *Lactobacillus* are common occupants of humans’ digestive tract that support the preservation of a balanced microbial environment that benefits host health. In this study, the unique lactic acid bacterium strain *Limosilactobacillus fermentum* U-21, which was isolated from the feces of a healthy human, was examined for its metabolite profile in order to compare it to that of the strain *L. fermentum* 279, which does not have antioxidant (AO) capabilities. By using GC × GC−MS, the metabolite fingerprint of each strain was identified, and the data were then subjected to multivariate bioinformatics analysis. The *L. fermentum* U-21 strain has previously been shown to possess distinctive antioxidant properties in in vivo and in vitro studies, positioning it as a drug candidate for the treatment of Parkinsonism. The production of multiple distinct compounds is shown by the metabolite analysis, demonstrating the unique characteristics of the *L. fermentum* U-21 strain. According to reports, some of the *L. fermentum* U-21 metabolites found in this study have health-promoting properties. The GC × GC−MS-based metabolomic tests defined strain *L. fermentum* U-21 as a potential postbiotic with significant antioxidant potential.

## 1. Introduction

Advanced multiomics technologies, which are largely focused on the universal detection of genes (genomics), mRNA (transcriptomics), proteins (proteomics), and metabolites (metabolomics) in a biological sample, have helped enhance studies on the gut microbiota. Multiomics technology enables researchers to study the gut microbiome with a better understanding of how the host and microbial community interact. The most recent and quickly developed branch of omics is metabolomics, which provides information about the biological and metabolic processes of microorganisms [1]. Metabolomics technologies allow studying molecular mechanisms of host-bacterial interactions and the mechanisms by which the hosts of the microbiota affect the host’s immunological status [2].

Researchers’ interest in *Lactobacillus* species’ metabolomics analysis has grown because these bacteria are the most common occupants of the gastrointestinal tracts of animals, including humans [3,4,5]. Lactobacillus is a species in the gut microbiome that is a natural source of postbiotics because of the active synthesis of biologically active compounds and the bidirectional communication with the host organism [6,7,8,9,10]. As a result, *Lactobacillus* can affect the immune status and antioxidant (AO) status of the host. Postbiotics are metabolites or components produced by microbiota that have an impact on human health [11]. Antioxidant properties of many *Lactobacillus* species have already been confirmed by many in vivo and in vitro studies [12,13,14,15]. The *Limosilactobacillus fermentum* U-21 strain used in this study is a potential pharmabiotic. The strains were selected previously based on their ability to reduce oxidative stress in vitro on a bioluminescent test system based on Escherichia coli MG1655 strains carrying plasmids encoding luminescent biosensors pSoxS-lux and pKatG-lux inducible by superoxide anion and hydrogen peroxide [13]. The antioxidant properties of the strain were next confirmed using in vivo and in vitro methods. In particular, *L. fermentum* U-21 prolonged the life by 25% of the nematode *C. elegans* under paraquat-induced oxidative stress [14]. *L. fermentum* U-21 stopped the degenerative alterations in internal organs [8] and the degeneration of brain dopaminergic neurons in a mouse model of paraquat-induced Parkinsonism [14]. The twenty-nine genes related to the antioxidant properties, which are involved in the thioredoxin complex, as well as heavy metal metabolism and transport, were found in the genome of the *L. fermentum* U-21 strain [16]. Comparative analysis of the proteins in the culture fluid of the *L. fermentum* U-21 strain and *L. fermentum* 279 strain, which does not exhibit AO properties [13,14], revealed significant differences associated with LysM and ATP-binding Clp protease [16]. ATP-binding Clp protease may be essential for the refolding of misfolded proteins caused by oxidative stress [17,18]. Both of the L. fermentum strains under study have the LysM protein, which can bind to peptidoglycan and affect the generation of humoral immune components [19]. The *L. fermentum* U-21 strain, however, has the short serine-threonine insertion [16]. The gathered data have shown that *L. fermentum* U-21 has special antioxidant characteristics, making it a promising candidate as a postbiotic [8,9,12].

In this study, we examine the metabolomic profile of the *L. fermentum* U-21 strain—candidate for a postbiotic—using an untargeted metabolomic technique. In addition, we compare the metabolites and related metabolic pathways among the *L. fermentum* U-21 strain, which demonstrates high AO properties, and the *L. fermentum* 279 strain, which does not exhibit AO properties. The main purpose of this study was to identify the unique metabolites of *L. fermentum* U-21 compared to *L. fermentum* 279, which does not exhibit AO properties.

## 2. Materials and Methods

### 2.1. Bacterial Strains

We used *L. fermentum* U-21 and *L. fermentum* 279 in this investigation, both of which were isolated from the bodies of people who lived in the Central European region of the Russian Federation. Both strains were preserved at the collection of the laboratory of genetics of microorganisms, Vavilov Institute of General Genetics Russian Academy of Sciences. The *L. fermentum* U-21 and 279 strains were characterized (Table 1).

### 2.2. Method, Conditions, and Media for Strain Propagation

The *L. fermentum* U-21 and *L. fermentum* 279 strains were both grown on the MRS media (Himedia) under partially anaerobic conditions (in a desiccator where oxygen was burned up by burning a candle). The MRS media contained 10.0 g/L proteosopeptone, 10.0 g/L meat extract, 5.0 g/L yeast extract, 20.0 g/L glucose, 1.0 g/L polysorbate 80, 2.0 g/L ammonium citrate, 5.0 g/L sodium acetate, 0.1 g/L magnesium sulfate, 0.05 g/L manganese sulfate, and 2.0 g/L dibasic potassium phosphate (pH 6.5 at 25 °C). For metabolomic analysis, the bacterial cultures were grown to the stationary growth phase (OD600 = 2.5). The cultivation temperature was 37 °C.

### 2.3. Phase Separation

Culture fluid of *L. fermentum* U-21 and *L. fermentum* 279 were prepared using the methanol-chloroform method with some modifications. All stages of selection pass at 4 °C. Bacterial cells were separated from the culture liquid at 7000× *g* for 10 min at 4 °C. After centrifugation, the culture liquids were filtered through a PES membrane (0.22 µm). The 40 mL of the supernatant was concentrated using Amicon Ultra-4 5000 MWCO. For further phase separation we selected 200 µL. Phase of culture fluid was separated by addition of 1 volume of methanol (*v*/*v*, 20 °C) and 2 volumes of chloroform, vortexed, and incubated for 20 min on ice. After incubation, we added 0.75 volume of Milli-Q water and incubated for 10 min on ice. After centrifugation at 8000× *g* for 10 min at 4 °C for phase separation, the aqueous phase was transferred to a 1.5 mL microcentrifuge tube. The methanol-chloroform method was conducted again for the upper lipid-containing phase. All samples, including extracellular vesicles, the aqueous, and lipid-containing phases, were evaporated using a SpeedVac concentrator.

### 2.4. Extracellular Vesicles Preparation and Physicochemical Analysis

Bacterial cells were separated from the culture liquid at 7000× *g* for 10 min at 4 °C. After centrifugation, the culture liquids were filtered through a PES membrane (0.22 µm). The 60 mL of supernatant was used to isolate the extracellular vesicles using ultracentrifugation at 260,000× *g* for 1 h 40 min at 4 °C and resuspended in phosphate-buffered saline (PBS, pH 7.4). Using a Qubit 3.0 fluorometer, the DNA, RNA, and protein quantities in the extracellular vesicles were quantified (Life Technologies, Grand Island, NY, USA). Extracellular vesicles in PBS (pH 7.4) suspension were deposited on a carbon film-coated copper grid treated by glow discharge and then stained with 1% uranyl for 3 min in an acetate solution for negative staining. Using the JEM-1400 (JEOL Inc., Tokyo, Japan) transmission electron microscope, the prepared grid was examined.

### 2.5. Extraction of Metabolites

Further sample preparation was carried out according to the protocol of O. Fiehn [20]. Then the metabolites were extracted with a mixture of isopropanol, acetonitrile, and water (3:3:2, *v*/*v*/*v*). An aliquot was evaporated to dryness, and then redissolved in a mixture of acetonitrile and water (1:1, *v*/*v*); the supernatant was evaporated to dryness.

### 2.6. Derivatization of Metabolites

In total, 10 µL of 20 mg/mL methoxyamine pyridine hydrochloride was added for derivatization, and the mixture was then shaken vigorously for 1.5 h at 30 °C. The samples were then further derivatized by adding 91 µL of a mixture of MSTFA and FAME to each sample, which was then incubated at 37 °C for 30 min on a thermal shaker. For GC × GC − MS analysis, the derivatized samples were put in a sample bottle.

### 2.7. GC × GC-MS Analysis

The GC-MS system consisted of an Agilent 7890/M780EI gas chromatograph (GC, Agilent Technologies, Santa Clara, CA, USA) coupled with a PERSEE mass spectrometer (MS, LECO, St. Joseph, MI, USA) and an L-PAL3 autosampler (PAL Systems, Zwingen, Switzerland). A sample of 1.0 µL was injected into a Restek Rxi-5Sil MS capillary column (30 m in length, Restek, Bellefonte, PA, USA) with a split ratio of 100:1 or 50:1. The column of the second dimension—Restek Rxi-17Sil MS (Restek, Bellefonte, PA, USA)—was 3 m 1in length. After sample preparation, a 70-eV electron beam was used to heat the electron impact (EI) ion source to 250 °C. Then, the solvent delay was 350 s, and the mass spectrometer transfer line temperature was 280 °C. The flow rate of the helium carrier gas was fixed to 1.0 mL/min. Following a 1 min hold at 60 °C, the GC oven ramped up to 280 °C at a rate of 10 °C/min. With a speed of 200 specters per second, masses were obtained in full scan mode spanning the range of 35 to 700 *m*/*z*. The chromato-mass spectrometer was controlled using the ChromaTOF software (v. 5.51, LECO, St. Joseph, MI, USA).

### 2.8. Data analysis

The resulting spectrum files were processed in ChromaTOF (v. 5.51, LECO, St. Joseph, MI, USA) for deconvolution, peak selection, alignment, and search in the primary database. Metabolites were identified based on mass spectra and retention times from the National Institute of Standards and Technology (NIST) libraries, the Mainlib and Feign libraries, and the National Institutes of Health (NIH) public repository.

We used ChromaTOF Tile v.1.01 (LECO, St. Joseph, MI, USA) to reduce the multidimensionality of experimental data based on the Fisher coefficient and identify significantly different chemicals in the culture liquids. The processing principle of ChromaTOF Tile v.1.01 is the comparison of two matching sections of the chromatogram (so-called tiles) and highlighting the low and high levels. The size of the studied cells was 3 *×* 24 in modulation and spectral measurements, respectively. Only results with a signal-to-noise ratio greater than 70 were counted. The range of analyzed masses was limited from *m*/*z* = 85 to *m*/*z* = 700. Identification was performed using the NIST database for mass spectra and retention indices (mainlib, replib) and the Leco-Fiehn rtx5 library. The matches with a direct and reverse similarity of more than 700 were considered significant.

Values with an F-ratio (intra-/inter-class variation ratio) greater than 5 were counted. A high F-ratio is predicted for a metabolite that varies between classes but is consistent within a class (low inter-class variation). The samples with a high F-ratio will be assigned to a different class (with three technical repeats and two controls of blank and pure methanol). The data were processed using the Python package, Pandas. N-Trimethylsilyl-N-Methyl Trifluoroacetamide (MSTFA), as well as its derivatives, were not taken into account. Duplicated metabolites were not counted. The samples were classified by the names of the metabolites. The metabolites were sorted by strains and fractions. Metabolite Set Enrichment Analysis (MSEA) was performed to distribute metabolites along metabolic pathways using the Metaboanalyst (https://www.metaboanalyst.ca/, accessed on 10 January 2023).

## 3. Results

### 3.1. Characteristic of the Metabolite Profile of the Culture Supernatant

Untargeted metabolite profiles of the samples from the *L. fermentum* U-21 strain and the strain *L. fermentum* 279, which does not exhibit AO properties, identified 681 metabolites in the aqueous phase of *L. fermentum* U-21 and 98 in the lipid-containing phase. Altogether, 257 metabolites of the total metabolite amount in the aqueous phase and 60 metabolites in the lipid-containing phase appeared to be the unique metabolites of *L. fermentum* U-21 in comparison with *L. fermentum* 279 (Figure 1).

The main classes of compounds detected included primary sugars, organic acids, amino acids in the aqueous phase and mostly fatty acids, and lipids in the lipid-containing phase. All common amino acids, except arginine, were detected in the aqueous phase of *L. fermentum* U-21; in the lipid-containing phase, alanine, asparagine, threonine, proline, and tryptophan were detected. There were nineteen organic acids in the aqueous phase and eight in the lipid-containing phase.

Twenty-four metabolites were reported in the literature to possess health-improving properties, of which six metabolites were amino acids. The remaining 18 metabolites were vitamin B3, vitamin B5, vitamin B6, L-Dopa, dopamine, noradrenaline, adrenaline, and GHB, which belong to different classes such as vitamins and neurotransmitters. Full, detailed results for the detected metabolites are shown in Table 2, Table 3 and Table 4 below and in Appendix A.

The metabolites were separated by metabolic pathways using the Metabolite Set Enrichment Analysis (MSEA). Figure 2 and Figure 3 show the MetaboAnalyst 5.0 pathway enrichment analysis used to evaluate the metabolic pathways most implicated in the results.

### 3.2. Physicochemical Characterization of L. fermentum U-21 Extracellular Vesicles

We looked for extracellular vesicles with spherical shapes (20–200 nm) in the *L. fermentum* U-21 culture supernatant. Transmission electron microscopy (TEM) was used to detect the presence of spherical structures (extracellular vesicles) in the ultracentrifugation of the bacterial culture supernatant. We verified the presence of structures with an approximate diameter of 80 nm (red arrow) in the *L. fermentum* U-21 ultracentrifugation residues, as shown in Figure 4a. The results of TEM observations shown in Figure 4a exposed that *L. fermentum* U-21 generated vesicle-like structures (approximately 80 nm).

Additionally, protein (820.7 ng/µL) and RNA (81.9 ng/µL) were present in *L. fermentum* U-21 extracellular vesicles, whereas DNA (21.6 ng/µL) was rarely detected (Figure 4b). This result is similar to what was seen for *L. paracasei* [21]. To summarize, some lactobacilli release extracellular vesicles into culture supernatants. The extracellular vesicles of *L. fermentum* U-21 appear to have a cytoplasmic membrane and mainly contain bacterial proteins.

### 3.3. Isolation and Characteristic of the Metabolite Profile of the Extracellular Vesicles

In the extracellular vesicles of *L. fermentum* U-21, 360 metabolites were discovered, 82 of which are unique in comparison with *L. fermentum* 279 (Figure 5 and Appendix A).

In the extracellular vesicles, forty-eight amino acids and their derivatives were found, six of which are unique to *L. fermentum* U-21: gamma-aminobutyric acid (GABA), cycloleucine, 3-hydroxy-tyrosine, alpha-methyl-L-tyrosine, aminomalonic acid, and alanylglycine. Extracellular vesicles included twenty-six sugars and carbohydrate conjugates, four of which are unique to *L. fermentum* U-21: 2,3,4-trihydroxybutyric acid, 2-keto-D-gluconic acid, *α*-D-mannopyranose, and arabinofuranose. There was also 2-keto-D-gluconic acid that is the key precursor for the synthesis of vitamin C detected. Nine metabolites were reported in the literature as bioactive compounds found in the extracellular vesicles, including GABA, adrenalin, dopamine, noradrenaline, melatonin, indole-3-acetamide, L-Norvaline, 4-Hydroxyphenyllactic acid with antioxidant properties, and penicillamine with anti-inflammatory properties (Table 5).

The sorting of metabolites by their metabolic pathways is performed with the metabolite set enrichment analysis (MSEA). Figure 6 shows results of the pathway enrichment analysis in MetaboAnalyst 5.0, which allow understanding of the main metabolic pathways.

## 4. Discussion

Host health can be influenced by the human gut microbiota, which can have different microbial compositions and biochemical functions, so prebiotics, probiotics, and postbiotics have attracted the attention of scientists as a viable way to change the intestinal microbiome. A healthy GIT can be supported by probiotics, which are live microorganisms that, when administered in adequate amounts, confer a health benefit on the host [22] and typically consist of *Lactobacillus* and *Bifidobacterium* species. However, the postbiotic is a relatively new concept in medicine and science. Despite the fact that the idea of postbiotics is raised from the concept of producing the secretion of various metabolites with beneficial effects by the microbiota [23], the useful properties of postbiotics and their bioactivities, which determine these properties, are still unknown or unclear. In our laboratory, sequencing of the genome of the *L. fermentum* U-21 strain was previously carried out and 29 genes potentially determining its antioxidant activity were annotated in its genome [16]. In the same work, genes, whose expression increased in response to exposure to hydrogen peroxide, were identified [16].

This study aims to characterize an *L. fermentum* U-21 species as a potential postbiotic and identify distinct metabolites that determine the health-improving unique properties of *L. fermentum* U-21 by comparing the metabolic profiles of two Lactobacillus strains, one studied strain, *L. fermentum* U-21, and another strain, *L. fermentum* 279, which does not exhibit high AO properties.

While analyzing the untargeted metabolomic profile, a total of 1144 metabolites in the *L. fermentum* U-21′s cultural fluid were identified: 681 metabolites of them in the aqueous phase, 98 in the lipid-containing phase, and 365 in the extracellular vesicles, as amounts was expected from previous work [24]. While comparing the metabolites of the studied *L. fermentum* U-21 strain with the *L. fermentum* 279, which does not exhibit high AO properties, 257 metabolites in the aqueous phase, 60 in the lipid-containing phase, and 82 in extracellular vesicles appeared to be unique for *L. fermentum* U-21. In addition, we identified that *L. fermentum* U-21 produced GABA, which is a nonprotein amino acid and a major inhibitory neurotransmitter in the mammalian central nervous system [25]. Many studies revealed considerable roles of GABA in different process such as modulating synaptic transmission, promoting neuronal development and relaxation, and preventing sleeplessness and depression [26,27,28,29]. Moreover, a recent study found that GABA has antihypertension, antidiabetes, anticancer, antioxidant, anti-inflammation, antimicrobial, and antiallergy properties [30]. However, the GABA synthesis gene has not been found in the genome of *L. fermentum* U-21, so we suppose that another metabolic pathway of GABA production is in this strain. Other neurotransmitters such as adrenaline, dopamine, noradrenaline, glycine, and l-DOPA were identified to be produced by *L. fermentum* U-21. An interesting fact is that a unique neurotransmitter found in *L. fermentum* U-21 but not *L. fermentum* 279 is l-DOPA. Since l-DOPA is the precursor of the neurotransmitters (dopamine, noradrenaline, and adrenaline), it is used in the therapeutic treatment of Parkinson’s disease and dopamine-responsive dystonia in medicine [31]. The culture supernatant contained a number of short-chain fatty acids, including acetic acid, propanoic acid, and butanoic acid. These saturated aliphatic organics acids regulate the immune system response [32]. In particular, butyrate can inhibit histone deacetylases (HDAC) and the activation of nuclear factor kappa *β* (NF-k*β*) in macrophages, which contribute to the immune and inflammatory response [33,34]. Some *Lactobacillus* species can convert tryptophan to indole, indole-3-aldehyde, and indole-3-lactic acid, which are signaling molecules to regulate epithelial integrity, immune response, and gastrointestinal motility through intestinal receptors. Indole and indole’s derivative such as indolelactate, indole-3-acetamide, and indole-3-acetic acid produced by *L. fermentum* U-21 have been reported to display immunomodulatory benefits [35].

In our laboratory, the genes involved in the first of two glutathione, which is a critical endogenous antioxidant, synthesis steps and the transport of glutathione had been recently found in the genome of *L. fermentum* U-21 [9]. Whereas metabolomic analysis of the strain revealed pyroglutamic acid, ornithine, cysteine, glutamic acid, and glycine, from which glutathione is synthesized, and enhanced glutathione’s synthesis [36].

The quantity of biological active metabolites of the promising postbiotic *Lactobacillus* strain *L. fermentum* U-21 was shown by this study of untargeted metabolomic analysis. This strain was firstly selected in an *E. coli* bioluminescent test-system for high AO potential [13]. *L. fermentum* 279, in turn, did not show high AO properties on this test-system. Previous studies have proved the high antioxidant and neuromodulation properties of *L. fermentum* U-21 by in vivo and in vitro models [13,14]. Moreover, recently unique genes coding the proteins involved in reducing antioxidant stress such as genes of a transcriptional regulator and genes responsible for heavy metal chelation and transportation were found in the *L. fermentum* U-21 and absence in the genome of *L. fermentum* 279 [16].

However, the biological compounds of the *L. fermentum* U-21 strain, which are responsible for such health-improving effects, have not been identified yet. Many strain-specific metabolites that can benefit the health of the host were found in this study. The production of these metabolites, which have positive health effects, may be the cause of this strain’s postbiotic effect. Knowing particular metabolic pathways and the mechanisms of the generation of particular metabolites is helpful for understanding strain function in the GIT and host organism, as well as for understanding the standards for choosing particular postbiotics in the future. Complex omics studies, including metabolomics and proteomics analysis, and analysis of mRNA presence in vesicles of *L. fermentum* U-21 with regulatory function are the subject of our studies. Obviously, the impact of the complex metabolites, proteins, and mRNA contained in the vesicles and delivered to needful organs causes target effects of rare probiotic strains on specific diseases [37,38,39].

## 5. Conclusions

The unique metabolites produced by *L. fermentum* U-21, such as numerous organic acids, including fatty and amino acids, as well as a number of neurotransmitters and vitamins possess health-beneficial properties. Therefore, this strain can be used as an ingredient in pharmaceuticals and health food products, but further metabolomics studies are needed to precisely understand biologically active metabolites production’s mechanisms. *L. fermentum* U-21 and its extracellular vesicles may be a potential candidate for the production of postbiotics and health food products.

## Figures and Tables

**Figure 1 biotech-12-00039-f001:**
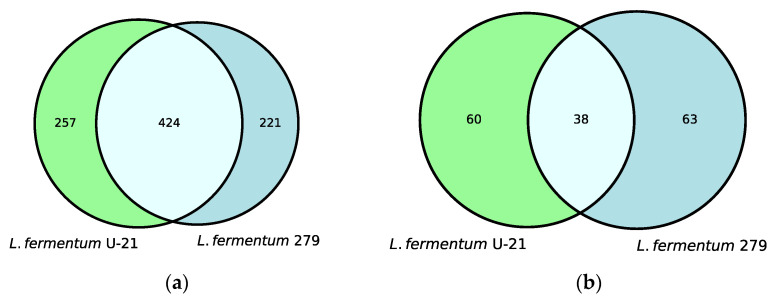
Venn diagrams show shared and unique metabolite features and their numbers for: (**a**) the aqueous phase of *L. fermentum* U-21 and *L. fermentum* 279; (**b**) the lipid-containing phase of *L. fermentum* U-21 and *L. fermentum* 279.

**Figure 2 biotech-12-00039-f002:**
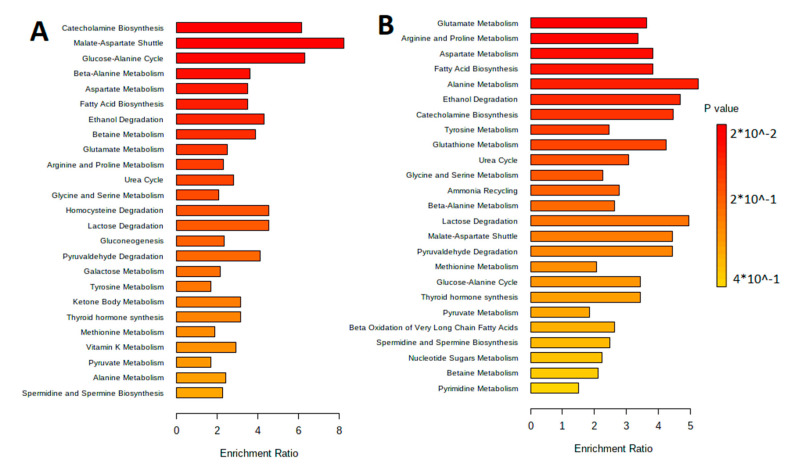
Pathway enrichment analysis for metabolites differed significantly between groups according to MetaboAnalyst 5.0 (*p* < 0.05). The number of hits inside a specific metabolic pathway divided by the anticipated number of hits yields the enrichment ratio. (**A**) The aqueous phase of *L. fermentum* U-21 culture supernatant. (**B**) The aqueous phase of *L. fermentum* 279′s culture supernatant.

**Figure 3 biotech-12-00039-f003:**
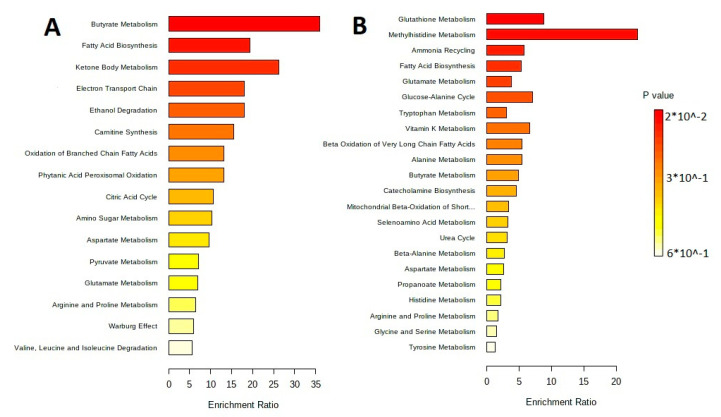
Pathway enrichment analysis for metabolites differed significantly between groups according to MetaboAnalyst 5.0 (*p* < 0.05). The number of hits inside a specific metabolic pathway divided by the anticipated number of hits yields the enrichment ratio. (**A**) The lipid-containing phase of *L. fermentum* U-21 culture supernatant. (**B**) The lipid-containing phase of *L. fermentum* 279′s culture supernatant.

**Figure 4 biotech-12-00039-f004:**
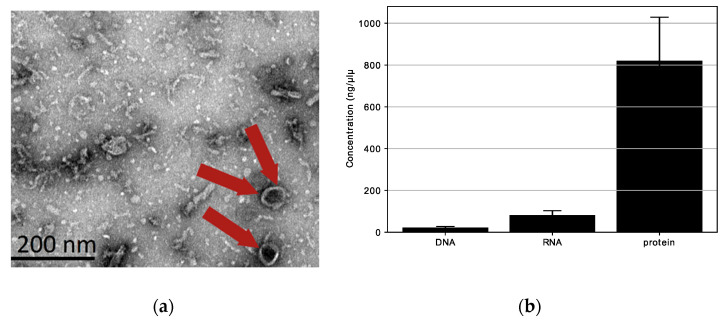
Characterization of extracellular vesicles (red arrows) produced by *L. fermentum* U-21: (**a**) TEM images of *L. fermentum* U-21 extracellular vesicles. Red arrows indicate extracellular vesicles. Scale bar, 200 nm; (**b**) DNA, RNA, and protein in *L. fermentum* U-21 extracellular vesicles. Means ± SD, n = 3.

**Figure 5 biotech-12-00039-f005:**
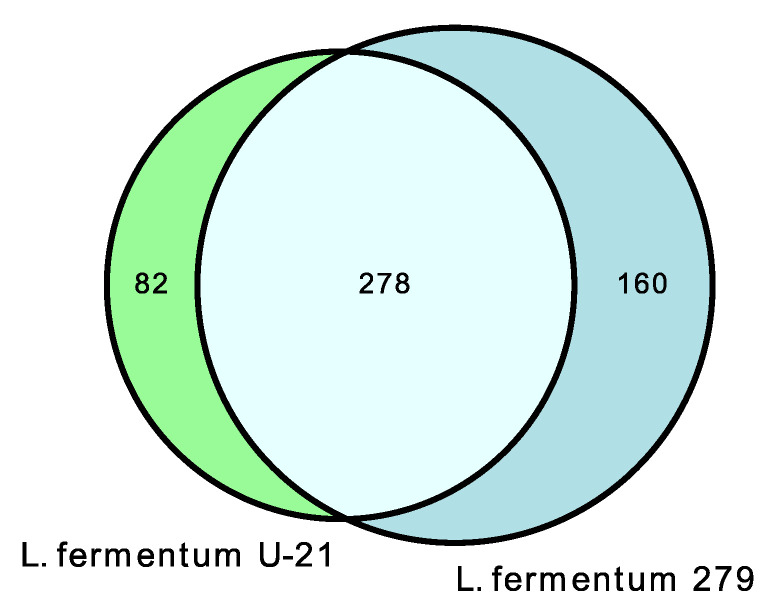
Venn diagrams show shared and unique metabolite features and their numbers for *L. fermentum* U-21 and *L. fermentum* 279 extracellular vesicles.

**Figure 6 biotech-12-00039-f006:**
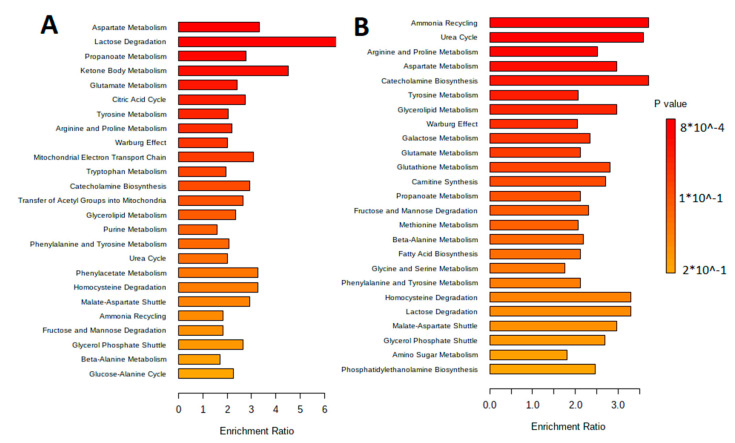
MetaboAnalyst 5.0 Pathway Enrichment Analysis for metabolites differed significantly between groups (*p* < 0.05). The enrichment ratio is calculated as the number of hits within a particular metabolic pathway divided by the expected number of hits. (**A**) Extracellular vesicles of *L. fermentum* U-21. (**B**) Extracellular vesicles of *L. fermentum* 279.

**Table 1 biotech-12-00039-t001:** Genetic characterization of strains.

Strain	BioSample ID	GenBank Sequence
*Limosilactobacillus fermentum* U-21	SAMN08290293	GCA_002869825.2
*Limosilactobacillus fermentum* 279	SAMN08014151	GCA_002794275.1

**Table 2 biotech-12-00039-t002:** Main classes of metabolites detected in the aqueous phase of culture supernatant of *L. fermentum* U-21.

Metabolite	Class	
Glycine (Gly) Alanine (Ala) Valine (Val) Leucine (Leu) Isoleucine (Ile) Aspartic acid (Asp) Asparagine (Asn) Glutamic acid (Glu) Glutamine (Gln) Serine (Ser) Threonine (Thr) Methionine (Met) Cysteine (Cys) Lysine (Lys) Histidine (His) Proline (Pro) Phenylalanine (Phe) Tyrosine (Tyr) Tryptophan (Trp) 2-Aminocaprylic acid Norleucine Homocysteine Homoserine Norvaline Ornithine *N*-Methyl-*α*-aminoisobutyric acid *β*-Alanine 5-Hydroxytryptophan Pyroglutamic acid Aceturic acid *α*-Methyl-l-tyrosine *N*,*N*-Dimethylglycine *N*-Acetyl-l-glutamic acid *N*-Acetylglutamine *N*-Acetyl-l-phenylalanine *N*-Acetyltyrosine	Amino acids and derivatives	
Carnitine Propanoic acid Cinnamic acid Succinic acid Formic acid Fumaric acid Maleic acid Malonic acid Phthalic acid Propanedioic acid Oxalic acid 2-Aminobutanoic acid 2-Hydroxybutyric acid 3-Hydroxybutyric acid Glycolic acid Lactic Acid Tartaric acid Methylcitric acid Acetic acid Hydracrylic acid Dodecanedioic acid	Organic acid	
Maltose Melibiose Mannobiose Cellobiose Lactose Turanose Xylofuranose 2-Amino-2-deoxyhexose Talose Erythrose Xylose Galactose Glucose Mannose Lyxose Ribose Threose Tagatose Fructose Psicose 2-Keto-gluconic acid Gluconic acid Mannonic acid Ribonic acid Arabinonic acid Galacturonic acid Gulonic acid Xylonic acid D-Erythro-Pentitol Mannitol Glycerol meso-Erythritol 1-Deoxypentitol Threitol Pentitol Arabinitol Erythritol Arabitol Arabinofuranose Dihydroxyacetone Levoglucosan Myo-Inositol Trehalose-6-phosphate Mannopyranose Glucose oxime Mannose oxime Arabinopyranose	Saccharides	
2-Hydroxyisocaproic acid 4-Hydroxybutanoic acid Nonanoic acid Butanoic acid	Fatty acids and analogues	

Underlined metabolites are unique for *L. fermentum* U-21.

**Table 3 biotech-12-00039-t003:** Main classes of metabolites detected in the lipid-containing phase of culture supernatant of *L. fermentum* U-21.

Metabolite	Class
Alanine (Ala) Asparagine (Asn) Threonine (Thr) Proline (Pro) Tryptophan (Trp) L-Norvaline 1-Aminocyclopentanecarboxylic Pyroglutamic acid 2-Hydroxy-3-methylbutyric acid 2-Methylalanine *N*-Methyl-l-glutamic acid	Amino acids and derivatives
Oxalic acid Phthalic acid Propanedioic acid Succinic acid Ritalinic acid 3-Hydroxyphenylacetic acid Acetic acid 4-Hydroxybenzoic acid	Organic acids
Lyxose Fructopyranose Mannitol meso-Erythritol	Saccharides
Heptadecanoic acid Undecanoic acid Butanoic acid 2-Hydroxyisocaproic acid Oleic acid	Fatty acids and analogues

Underlined metabolites are unique for *L. fermentum* U-21.

**Table 4 biotech-12-00039-t004:** Health beneficial compounds detected in the culture supernatant of *L. fermentum* U-21.

Metabolite	Class
Dopamine Histamine Noradrenaline Adrenaline GABA l-DOPA Glycine 4-Hydroxybutanoic acid (GHB)	Neurotransmitter
Indole-3-acetamide Indole Indolelactate Indole-3-ethanol	Indoles and derivatives
Tartaric acid *β*-Alanine Carnitine	Antioxidant
Cinnamic acid L-Norvaline	Anti-inflammatory
5-Hydroxytryptophan	Antidepressant
Pyridoxine	Vitamin B6
Niacin Niacinamide	Vitamin B3
Pantothenic acid	Vitamin B5
Acetamide	Antimicrobial

Underlined metabolites are unique for *L. fermentum* U-21.

**Table 5 biotech-12-00039-t005:** Health beneficial compounds detected in the culture supernatant of *L. fermentum* U-21.

Metabolite	Class
Adrenaline GABA	Neurotransmitter
5-Hydroxytryptophan	Antidepressant
Timonacic 5-Methoxytryptamine (5-MT)	Antioxidant
Indole-3-acetamide	Indoles and derivative
Melatonin	Hormone
2-keto-D-gluconic acid	Vitamin C precursor
L-Norvaline Penicillamine	Anti-inflammatory
3-Phelyllactic acid	Antimicrobial

Underlined metabolites are unique for *L. fermentum* U-21.

## Data Availability

The analyzed open access materials are available online in issues on journal websites. The data are available in the Appendix A.

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
