# Peer review of "Metabolites Potentially Determine the High Antioxidant Properties of Limosilactobacillus fermentum U-21"

_biotech, 2023, doi:10.3390/biotech12020039_

Round 1
Reviewer 1 Report
In the Bacterial strains part, you need to do genetic identification for L. fermentum U-21 and L. fermentum 279.
in the supplement materials, you need to supply the shared metabolites for L. fermentum U-21 and L. fermentum 279.
in Table 1-3, these metabolites are only for L. fermentum U-21. you should also supply the metabolites for L. fermentum 279.
What puzzles me is this there is few or no results for L. fermentum 279.
you should make all the fiugres clear.
in isolation and characterisitic of the metabolite profile of the extracellular vesicles, you should supply the shared 283 metabolites between L. fermentum U-21 and L. fermentum 279
in discussion part, you should discuss why the L. fermentum 279 did not prouduce antioxidant substances.
Author Response
We would like to thank you for the constructive review of our manuscript. The provided comments are very helpful and we have revised the manuscript as suggested. Please find below the point-by-point response to the reviewers’ comments.
The manuscript has been checked and corrected by a professional native English translator. Thank you very much for your valuable advice. Now the English in the text is completely correct. It can be published.
“In the Bacterial strains part, you need to do genetic identification for L. fermentum U-21 and L. fermentum 279.”
Response: In the Bacterial strains part, we added table, containing genetic information. The annotation of both strains was done by the use of NCBI Prokaryotic Genome Annotation Pipeline (PGAP) (line 81).
“In the supplement materials, you need to supply the shared metabolites for L. fermentum U-21 and L. fermentum 279.”
Response: We added the data about the shared metabolites of L.fermentum U-21 and 279 to the supplement materials (Table 1, 3)
“In Table 1-3, these metabolites are only for L. fermentum U-21. You should also supply the metabolites for L. fermentum 279.”
Response: Thank you for this note. We have added a data table, containing information about L. fermentum 279 metabolites to the supplement materials of the article (Table 2, 4).
“You should make all the figures clear.”
Response: We changed Figure 4 (a) to the updated one (line 231). Also, the figures were made clear.
“In isolation and characterisitic of the metabolite profile of the extracellular vesicles, you should supply the shared 283 metabolites between L. fermentum U-21 and L. fermentum 279”
Response: The table, containing the information about main classes of the extracellular vesicles shared metabolite was added to the supplementary materials (Table 5). We also analyze the table again and clarify the quantity.
“In discussion part, you should discuss why the L. fermentum 279 did not produce antioxidant substances.”
Response: The Limosilactobacillus fermentum U-21 have studied in our laboratory many years. The strain were selected previously based on their ability to reduce oxidative stress in vitro on a bioluminescent test system based on Escherichia coli MG1655 strains carrying plasmids en-coding luminescent biosensors pSoxS-lux and pKatG-lux inducible by superoxide anion and hydrogen peroxide (Marsova, M.; Abilev, S.; Poluektova, E.; Danilenko, V. A bioluminescent test system reveals valuable antioxidant properties of lactobacillus strains from human microbiota. World Journal of Microbiology and Biotechnology 2018, 34, 1–9.). The L. fermentum 279 did not show high AO activity in these bioluminescent test-system.The antioxidant properties of strain was next confirmed using in vivo and in vitro. (Marsova, M.; Poluektova, E.; Odorskaya, M.; Ambaryan, A.; Revishchin, A.; Pavlova, G.; Danilenko, V. Protective effects of Lactobacillus fermentum U-21 against paraquat-induced oxidative stress in Caenorhabditis elegans and mouse models. World Journal of Microbiology and Biotechnology 2020, 36, 1–10.). Moreover, comparative genomic analysis of L. fermentum U-21 and 279 revealed unique genes involved in reducing oxidant stress. We added in the Discussion part explanation why L. fermentum 279 did not produce antioxidant substances: “This strain was firstly selected in E.coli bioluminescent test-system for high AO potential [13]. L. fermentum 279, in turn, did not show high AO properties on this test-system. Previous studies have proved the high antioxidant and neuromodulation properties of L. fermentum U-21 by in vivo and in vitro models [13, 14]. Morover, recently unique genes coding the proteins that involve in reducing antioxidant stress such as genes of a transcriptional regulator and genes responsible for heavy metals chelation and transportation were found in the L.fermentum U-21 and absence in the genome of L.fermentum 279” (line 331-337).
Reviewer 2 Report
The paper “GC-MS Metabolomics Characteristics of Limosilactobacillus fermentum U-21” approach an interesting and actual topic, namely that of probiotics and postbiotics. In general, the results are presented clearly, and the discussions are related to the other research carried out on these strains of lactobacilli. However, some changes should be made to the article in order to be published. The suggestions can be found in the attached document.

Author Response
The manuscript has been checked and corrected by a professional native English translator. Thank you very much for your valuable advice. Now the English in the text is completely correct. It can be published.
“183-186 This phrase that presents the total number of metabolites in the aqueous and lipid phase can be confusing for the reader, I recommend rewriting it for greater clarity.”
Response: Thank you for pointing this out. We have rewritten this phrase (line 174-179).
“199-201 – “The remaining 18 metabolites, which were vitamins, indoles, neurotransmitters, and organic acids include vitamin B3, vitamin B5, vitamin B6, L-Dopa, dopamine, noradrenaline, adrenaline, GHB”- it seems to be an unfinished sentence, please reformulate it in context”
Response: Thanks a lot. The sentence was corrected (line 192-194).
“What technique was used to obtain the results presented in Figure 4.b - The content of DNA, RNA, and protein in L. fermentum U-21 extracellular vesicles? ”
Response: The DNA, RNA and protein concentration of the extracellular vesicles were measured by the Qubit 3.0 fluorometer. We followed the same approach as in the article (Kurata, A., Kiyohara, S., Imai, T. et al. Characterization of extracellular vesicles from Lactiplantibacillus plantarum. Sci Rep 12, 13330 (2022). https://doi.org/10.1038/s41598-022-17629-7]).
“244 – repeated word.”
Response: The error was corrected as suggested (line 237).
“340 - The metabolites produced by L. fermentum U-21 are very beneficial. I do not consider this to be the main conclusion of the study, so I recommend rewrittng the conclusions part.”
Response: Thank you for pointing this out. The conclusion part was reformulated (line 351-357).
We have also made some minor style and typos corrections. I hope the revised manuscript is now acceptable for publication.
Thank you for your consideration.
Round 2
Reviewer 1 Report
In this paper, the author used untargeted metabolomic approach to identify the metabolites of the L. fermentum U-21 strain and the L. fermentum 279 strain, they found that the U-21 strain can produce some antioxidants, that is interesting to some extent.
Author Response
Dear Reviewer,
Thank you for your recommendations to improve our manuscript.
We agree with all your comments.
The manuscript have been revised.
Reviewer 2 Report
The authors made significant improvements to the writing of the article and followed the reviewer's recommendations.
Author Response

(The authors gave the same response as above.)
